# Facilitators for and Barriers to the Implementation of Performance Accountability Mechanisms for Quality Improvement in the Delivery of Maternal Health Services in a District Hospital in Pwani Region, Tanzania

**DOI:** 10.3390/ijerph20146366

**Published:** 2023-07-14

**Authors:** Francis August, Tumaini Mwita Nyamhanga, Deodatus Conatus Vitalis Kakoko, Nathanael Shauri Sirili, Gasto Msoffee Frumence

**Affiliations:** 1Department of Development Studies, School of Public Health and Social Sciences, Muhimbili University of Health and Allied Sciences, Dar es Salaam P.O. Box 65015, Tanzania; tnyamhanga69@gmail.com (T.M.N.); drnsirili@gmail.com (N.S.S.); gasto.frumence65@gmail.com (G.M.F.); 2Department of Behavioral Sciences, School of Public Health and Social Sciences, Muhimbili University of Health and Allied Sciences, Dar es Salaam P.O. Box 65015, Tanzania; deodatuskakoko@gmail.com

**Keywords:** accountability mechanisms, maternal health, barriers, facilitators quality improvement

## Abstract

Tanzania experiences a burden of maternal mortality and morbidity. Despite the efforts to institute accountability mechanisms, little is known about quality improvement in the delivery of maternal health services. This study aimed at exploring barriers and facilitators to enforcing performance accountability mechanisms for quality improvement in maternal health services. A case study design was used to conduct semi-structured interviews with thirteen key informants. Data were analyzed using thematic analyses. The findings were linked to two main performance accountability mechanisms: maternal and perinatal death reviews (MPDRs) and monitoring and evaluation (M&E). Prioritization of the maternal health agenda by the government and the presence of maternal death review committees were the main facilitators for MPDRs, while negligence, inadequate follow-up, poor record-keeping, and delays were the main barriers facing MPDRs. M&E was facilitated by the availability of health management information systems, day-to-day ward rounds, online ordering of medicines, and the use of biometrics. Non-use of data for decision-making, supervision being performed on an ad hoc basis, and inadequate health workforce were the main barriers to M&E. The findings underscore that barriers to the performance accountability mechanisms are systemic and account for limited effectiveness in the improvement of quality of care.

## 1. Introduction

Globally, as of 2020, maternal mortality ratios (MMRs) were estimated at 223 deaths per 100,000 live births, denoting a 38% reduction since the year 2000. Sub-Saharan Africa (SSA) accounted for 56% of the global maternal mortality ratio [1]. One of the major contributors to the high MMR in SSA is the non-responsiveness of the health systems [2,3,4]. As for the rest of SSA, the MMR in Tanzania is unacceptably high and stood at 556 maternal deaths per 100,000 live births in 2016 [5]. The poor response of the health system manifests itself as poor quality of maternal newborn health services [6,7,8,9]. Strengthening accountability in health systems has been seen as instrumental in turning the tide of the MMR. Accountability refers to having the obligation to answer questions regarding decisions and/or actions and the imposition of sanctions for failure to comply and/or to engage in appropriate action. There are three categories of accountability, namely: financial, democratic, and performance [10]. This study is based on performance accountability mechanisms, namely maternal and perinatal death reviews together with monitoring and evaluation. Performance accountability is defined as demonstrating and accounting for performance in light of agreed targets in terms of services, outputs, and outcomes. On the basis of this definition, performance accountability mechanisms include maternal and perinatal death surveillance reviews, professional norms, standards and bodies, and health facility committees, together with monitoring and evaluation. This paper has solely focused on only two mechanisms, namely maternal and perinatal death surveillance reviews together with monitoring and evaluation [1]. The MPDSR (Maternal and Perinatal Death Surveillance and Response) involves constant surveillance to quantify deaths (Surveillance), the review of circumstances to ascertain causes and contributing factors, and then making recommendations to address the causes (Response). In Tanzania, it started in 2013. This emphasizes the response, which is making recommendations to address the causes and contributing factors [1,2,3]. Furthermore, monitoring is involved with the ongoing systematic collection and analysis of data as the delivery of maternal and newborn care progresses, aiming at measuring progress toward the achievement of quality improvement objectives including better maternal and newborn outcomes. Evaluation refers to a process through which the hospital determines its success in improving maternal and newborn outcomes [11].

Although accountability in health systems has been seen as instrumental in turning the tide of the MMR, little attention is invested in strengthening accountability. Indeed, most of the studies conducted in the area are much more focused on the general health system domains, human resources for health [12,13,14], health financing [15,16], medical supply chain [17], and their contribution or failures to contribute to quality improvement in maternal health care. Furthermore, a systematic review [1] documented aspects of performance accountability such as context and mechanism of change, denoting that they are arbitrarily explored.

The findings from this study will shed light on the context-specific factors that not only support but also hinder the efficacious implementation of performance accountability mechanisms for maternal and child health services.

Despite the existence of performance accountability mechanisms, such as MPDSR, M&E system, clinical meetings, and ward rounds [4], the MMR and newborn mortality reduction are not being achieved in Tanzania, or they are inequitably achieved [18,19,20]. These mechanisms are implemented by the hospital management team (HMT). Studies exploring barriers to performance accountability have been limited in scope [21,22]. Firstly, most of these studies have been conducted outside Tanzania [23,24,25] and have had a predominant focus on enablers with relatively less attention on the barriers [19,22,26]. The few that focused on barriers to the implementation of performance accountability mechanisms explored staff-related limitations with less attention on the contextual constraints. Consequently, factors contributing to the insignificant reduction in the MMR and newborn mortality in Tanzania are not sufficiently understood. That is, it is not sufficiently clear why the existence of performance accountability mechanisms is not yielding the expected improvement in maternal and newborn outcomes. Therefore, this paper sought to explore facilitators and barriers to the implementation of performance accountability mechanisms for maternal and newborn health services, specifically on maternal and perinatal death reviews and monitoring and evaluation [27].

## 2. Materials and Methods

### 2.1. Study Design

We used a case study design [28] to explore the barriers and facilitators for enforcing performance accountability mechanisms for quality improvement in the delivery of maternal newborn and child health services in one of the district hospitals in the Pwani Region in Tanzania. A case study design was used in this study to allow an in-depth and comprehensive exploration of barriers and facilitators to the enforcement of selected performance accountability mechanisms in the real district hospital context.

### 2.2. Study Context

The health system in Tanzania performs in a decentralized manner [29], whereby the health care system is organized into three levels: the primary level (district hospital) where this study was situated, the secondary level (regional hospitals), and the national hospital. This study was conducted in one of the district hospitals in the Pwani Region in 2020. The region consists of nine district hospitals, namely: Mkuranga, Kisarawe, Kibiti, Rufiji, Bagamoyo, Mafia, Chalinze, Kibaha DC, and Kibaha TC.

### 2.3. Sampling Strategy and Recruitment of Study Participants

We purposefully selected thirteen key informants (KIs) based on their administrative positions in the hospital, which involved, among others, overseeing performance accountability in their respective units. Thus, we recruited a medical officer in charge; a hospital secretary; representatives in charge of nursing/midwifery, in charge of RCH, in charge of pharmacy, in charge of the laboratory, in charge of the labor ward, and in charge of theatre; a doctor in the labor ward, a doctor in the RCH, an anesthesiologist, a representative in charge of complaints, and doctor in theatre.

### 2.4. Data Collection Techniques

#### Key Informant Interviews

We interviewed key informants using a semi-structured interview (SSI) guide containing questions exploring the barriers to and facilitators for enforcing accountability mechanisms for the delivery of maternal newborn and child health services. The interview guide was developed in response to the research question: what are the facilitators for and barriers to the implementation of performance accountability mechanisms for the delivery of quality maternal health services in a district hospital? Therefore, the guide had questions about existing accountability mechanisms, how they are implemented, factors enabling their implementation, and challenges facing the implementation of the identified mechanisms. The questions were developed by the first author and reviewed by the other co-authors. The interviews were conducted in the informant’s office or in a room that was found convenient within the hospital, as chosen by the informant. Interviews lasted for approximately 30 to 60 min. The first author conducted the interviews while a research assistant, who accompanied the researcher, took field notes and managed the audio recorder. We conducted interviews and discussions in Kiswahili and audio-recorded the interviews.

### 2.5. Data Management and Analysis

First, we transcribed all the audiotaped data verbatim and then translated them from Kiswahili to English. Then we used an inductive thematic approach to analyze data [30,31,32,33]. To familiarize themselves with the data, all authors read the transcripts and field notes before commencing the analysis.

Six steps were followed including familiarization, coding, generating themes, reviewing themes, defining and naming themes, and writing up [34]. The first author and the second author developed a preliminary codebook concerning this study’s objective, which was to explore the facilitators for and barriers to implementing accountability mechanisms for enhancing quality improvement in maternal health care.

The first author and the second author coded meaningful units for the codes that represented facilitators and barriers, and they were coded more than once. We sorted codes into sub-themes and further into themes and made comparisons to check for similarities and differences. A more detailed scrutinization was carried out to review, identify, summarize, and retain patterns, similarities, differences, and emerging themes [35,36].

### 2.6. Trustworthiness of the Findings

To achieve trustworthiness in this study, three issues were taken into consideration, namely credibility, dependability, and transferability [37]. Credibility involves confidence in the truth of the study findings, and it was ensured using a prolonged engagement with the informants by building trust with them so that they could engage freely and openly together with deep discourse about the subject matter. The nature of the prolonged engagement in the data collection process meant that the researcher used different techniques such as paraphrasing the questions and using more probes during the interviews. Apart from the time spent during the interview, other techniques were included such as participant observation of the daily activities at the facility, including how the suggestion boxes operate, and looking at how patients had access to the available phone number for reporting complaints. Probing tactics were also used to uncover all the possible responses from the informants about the subject matter. All the researchers were conversant with the subject matter on the facilitators for and barriers to enforcing accountability in maternal care. Furthermore, the credibility of the methods was ensured by making sure that the sample selected was appropriate to the subject matter on the barriers to and facilitators for the implementation of accountability mechanisms.

Transferability has to do with how applicable the findings are in other settings with similar contextual factors [38,39], and this was ensured using a thorough description of the study setting, data collection techniques, informants, data analysis, and findings. Concerning dependability, each interview was considered a lesson for the subsequent interview, meaning that we learned the best practices to improve the interview process with a better comprehension of the phenomena in the implementation of accountability mechanisms [40,41].

Dependability refers to the ability of the researcher to account for changes in research over time, including data collection and analysis. With the aid of the interview guide, we ensured consistency and openness to the new insights in the open-ended questions. The insights obtained were taken into account in the subsequent interviews and analysis process [41,42].

### 2.7. Ethical Considerations

We obtained ethical clearance to undertake this study from the Muhimbili University of Health and Allied Sciences (MUHAS) Institutional Review Board (IRB) before data collection. Permission to collect data was obtained from the regional and local government authorities. We also sought informed consent from the participants, who were given consent forms to read and sign. For the protection of participants, we assured them of confidentiality, privacy, and anonymity. We conducted interviews in suitable places such as the offices of the respondents to ensure privacy and comfort for the participants during interviews. Lastly, we excluded names and other possible identifiers from the data set to ensure anonymity for the study participants.

## 3. Results

The findings are linked to two main performance accountability mechanisms, namely, maternal and perinatal death reviews and monitoring and evaluation. From the analysis, four themes and their respective sub-themes emerged, as shown in (Table 1).

### 3.1. Facilitators for Maternal and Perinatal Death Reviews

The analysis illustrated two themes, which include the prioritization of the maternal health agenda by the government and the availability of maternal death review committees at the facility.

### 3.2. Prioritization of the Maternal and Child Health Agenda by the Government

The participants pointed out that for quite some time, the government of Tanzania has been placing the agenda of women’s and children’s health on its list of priorities. Thus, implementing maternal and perinatal death reviews was in line with implementing the government directives. The respondents claimed that maternal death review reports were submitted to the district and regional commissioners for further follow-ups and implementation. This situation enabled the maternal and perianal death review process to be smooth.


*Ever since the fourth phase of government led by the former President Kikwete, issues of maternal and child health were taken as priority-political agenda by the ruling party. Even the fifth phase government led by the late President Magufuli spearheaded the agenda by putting more emphasis on the construction of facilities and procurement of medical supplies. Also, the regional and district commissioners were mandated to oversee issues of maternal mortality in their respective administrative areas. (KI_1)*


### 3.3. Perceived Availability of Active Committees for the Review of Maternal and Perinatal Mortality

The respondents pointed out that the facility had special committees composed of the medical officer in charge of the hospital and all the representatives in charge of each section. The informants reported that the duty of the committee was to make follow-ups, scrutinize all maternal and perinatal deaths that occurred at the hospital, and, thereafter, submit the report to the district commissioner.


*“There is a maternal death committee composed of all the in-charges of the hospital departments, i.e., the in-charge of nursing, in-charge of the labor ward, the in-charge of the RCH, and the in-charge of the pharmacy. The committee requires all those who were on duty to provide explanations to establish the cause of that particular death. This is in line with the government’s directive which indicates that when a maternal death occurs, it has to be audited within 24 h. Thus, when we receive a maternal death report or perinatal death, we respond very quickly and audit the death using our team, which consists of the in-charge of nursing, pharmacist, lab manager, and a doctor. We conduct the audit to establish the cause, particularly where the problem emerged from and identify a solution.” (KI_2)*


### 3.4. Barriers to Implementing Maternal and Perinatal Death Reviews

#### 3.4.1. Perceived Poor Documentation

The informants alluded that sometimes, medical records are poorly recorded/documented due to a lack of sufficient medical recording systems such as a computer server. The informants claimed that handwritten reports were subject to many inconsistencies and incompleteness and thus failed to indicate the action points for implementation. The participants added that with poor documentation, it becomes difficult to have a sufficient follow-up on past scenarios.


*“Our systems of operation are very challenging; we don’t have enough computers with servers to store the patient’s information including the maternal review reports. Most of our reports are handwritten which are full of errors and inconsistencies and stored in the shelves” KI_4*


#### 3.4.2. Frustrations Due to Inadequate Follow-Ups and Negligence on the Action Points Identified during Maternal Death Audits

It was reported that facilities suffer from ineffective implementation of the action points identified by review panels; this resulted in frustrations among the providers of maternal care, especially those who are in charge of the specific sections. Due to negligence among leaders, the respondents claimed that some do not fulfill their responsibilities such as making follow-ups on the implementation of recommendations arising from the reviews and convening regular meetings; rather, they wait until there is an incidence of maternal death, after which, they turn up. The informants alluded that the health facility lacks sophisticated mechanisms for keeping patients’ information. The informants claimed that handwritten reports were subject to many inconsistencies and completeness, and thus they failed to indicate the action points for implementation.


*Maternal death reviews are normally done as part of the routine process, however, the proposals, recommendations and action plans laid down by the committees are either delayed or not implemented at all. Not only that but also, these reviews are not undertaken on time... they may take a month or so to do the review, and in most cases recommendations appear to be out of context, reminding people about things they do not want to hear again! KI_5*


#### 3.4.3. Delays in Conducting the Reviews

The informants reported that there are delays in conducting maternal and perinatal reviews due to fear of blame. In addition, sometimes delays occur because the individual who was involved in the management of the diseased may hide some of the data/information or sometimes there is poor documentation of the information related to the deceased. The respondents alluded that delays in conducting reviews are an impediment to the implementation of MPDRs. It was also reported that delays happen because of fear of blame, that is, leaders fear taking action against the healthcare worker who was responsible for the deaths because they fear being blamed. This means that if responsible personnel perform reviews as early as required, it will be easy to uncover the reasons for deaths and propose actions against those responsible. However, when they delay conducting reviews, the implementation becomes difficult since some information might be missing from the files or some persons might have been transferred to other sections. Thus, it has become common that when matters are delayed, it is difficult to come up with strict action points against those who were responsible.


*It is very challenging when the doctor or nurse/midwife who managed the diseased person is involved in the review process. They always hide important information related to the diseased person or poorly document the information related to the maternal or perinatal death... this leads to unnecessary delays and compromises the whole process. KI_6*


#### 3.4.4. Facilitators for Monitoring and Evaluation

The facilitators for monitoring and evaluation were conceptualized as aspects that enhance the smooth implementation of this performance accountability mechanism. Accordingly, the analysis revealed three themes that fit as factors for facilitating the implementation of monitoring and evaluation. They include the availability of a health management information system, online ordering for medical supplies, and day-to-day implementation of ward rounds.

#### 3.4.5. Availability of a Health Management Information System

The informants reported that the most important enabling factor for implementing monitoring and evaluation interventions to manage and analyze aggregate information obtained from health facilities in the district and for enhancing quality improvement in maternal health services was the institutionalizing DHIS-2 (District Health Information System, version two, Tanzania). All the health-related data are entered into this system, thus making it easy to track service delivery at the facility. The context of this study implies the district hospital needs to have hospital-based data collection to capture hospital status and health outcomes and to monitor and evaluate performance by tracking indicators closely related to maternal and neonatal mortality. Such indicators include impact indicators such as the number of maternal and newborn deaths, the number of maternal deaths by cause, postpartum hemorrhage, sepsis, obstructed labor, eclampsia, unsafe abortion complications, HIV/AIDS (indirect), TB (indirect), malaria (indirect), and the number of neonatal deaths (in the hospital). The outcome indicators include the percentage of deliveries in the hospital, the number of direct obstetric complications by cause, postpartum hemorrhage obstructed labor eclampsia, unsafe abortion, complications, the number of cesarean sections performed, the number of newborns successfully resuscitated, and the number of infants breastfed within one hour of birth. The output indicators include the number of EmONC, signal functions performed, HIV tests conducted, and PMTCT (prevention of mother-to-child HIV transmission) services provided. The informants alluded that the existence of health information made it easy for the facility to have data that is useful for monitoring and evaluating day-to-day service delivery at the facility.


*‘The government has installed an electronic system (DHIS_2) District Health Information system (version two) which replaced MTUHA books which sometimes lacked complete data or sometimes contained lots of errors... data generated from this electronic system can be easy extracted and being used for planning, monitoring and evaluation of the hospital tasks.’ KI_8*


#### 3.4.6. Informative Online Ordering Systems for Medical Products Instituted by the MSD (Medical Stores Department)

The participants claimed that before the enactment of the online ordering/procurement platform, there used to be a paper-based system that was bureaucratic and led to unnecessary delays, and sometimes, the papers would get lost and were difficult to trace. It was also reported that requisitions were not channeled to MSD directly but could go to private vendors. With the online system for procurement, health facilities can openly press orders for the items they want and have access to trace them online if they have reached the MSD or not. The current procurement process that enables facilities to press orders for their medical supplies directly from the MSD has made it easy. The online system makes it easy to hold accountable careless pharmacists who fail to order the needed medications on time. This was reported to improve the M&E process because preceding M&E, reports used to document that most deaths were due to delays in the procurement of medical supplies.


*The current system of ordering/procuring medical products has been made simple as compared to the previous system, you can press your order online from MSD... when they are out of stock you order from the private vendors/suppliers… if the facility runs out of stock, it implies that the is some negligence based on the current ordering system KI_9*


#### 3.4.7. Day-to-Day Implementation of Ward Rounds

The informants reported that weekly major ward rounds constitute part and parcel of performance accountability (monitoring and evaluation). This is performed by a senior doctor or doctor in charge of the maternal and newborn units in the presence of subordinate staff doctors and nurses. The team goes around the wards, from bed to bed, checking on the hospitalized patients’ progress. After the ward rounds, steps are taken to fill the identified gaps, thereby strengthening the monitoring and evaluation of the health services to mothers and babies.


*“We do ward rounds. As doctors, we write prescriptions, but the nurses are the ones who perform or administer the treatment and other clinical procedures. The medical practitioner on duty has to oversee and check to see if the treatment procedures and drug administration are done as prescribed. We also interview patients regarding their treatment progress. After the ward rounds, we write reports or recommendations from what we observe or hear from patients.” KI_10*


#### 3.4.8. Barriers to Implementing Monitoring and Evaluation

Barriers to the implementation of monitoring and evaluation were conceptualized as aspects that hinder the implementation of this performance accountability mechanism. Accordingly, the analysis revealed three themes that fit as factors that hinder the implementation of monitoring and evaluation. They include the fragmented M&E system, supervision performed on an ad hoc basis, supply chain deficiencies, and human resources for health challenges.

#### 3.4.9. Supportive Supervision Performed on an Ad Hoc Basis

The respondents believed that monitoring and evaluation are enforced with regular supervision and monitoring of service providers and health administrators. However, they cautioned that surveillance and monitoring are not regularly performed unless there is an incident, such as a maternal death, and monitoring and evaluation are not rigorously implemented. This was well explained by one of the nurses working in the labor ward.


*Follow-up and supervision should be done regularly regardless. It happens that only when there is a maternal death then you see HMT members going around for supervision and interrogation … this is not a good practice … we need to institute regular supervision and feedback mechanisms that can be used to monitor the performance of supervisors (the in-charges of the sections … the watchdog must also be watched!.......”KI_12*


#### 3.4.10. Inadequate Human Resources with Skills in Monitoring and Evaluation

This was termed as a major setback against the adequate implementation of monitoring and evaluation. The studied district hospital had no specific personnel with the capacity to monitor and evaluate the delivery of maternal and newborn care. The lack of adequate human resources for health with the specified skills makes it difficult to monitor and evaluate tasks, especially for the delivery of maternal and born health services. The participants noted that given this challenge, M&E programs operate sub-optimally, leading to deficiencies in quality improvement for maternal care. Implementation of M&E is supposed to be carried out by well-trained and skilled experts in the subject matter. In Tanzania, the implementation of M&E is characterized by the use of HMIS, which involves the use of registers documenting information about patient visits, diagnosis and treatment outcomes, and health facilities. An aggregate summary of the forms is submitted to the district level to be entered into the computer software DHIS-2.


*“Another big problem we are experiencing is the shortage of labour force who are well equipped with monitoring and evaluation competencies/skills, the available workers with the specified skills are very few. This tends to compromise the M&E programme at the facility and leads to sub-optimal results in the delivery of maternal and newborn health services. The government’s initiative to remove ghost workers and the unqualified (those with fake certificates) has left us with a huge shortage of human resources for health including the removal of healthcare workers skilled in M&E” KI_13*


## 4. Discussion

This study sought to explore facilitators and barriers to implementing accountability mechanisms for enhancing quality improvement in the delivery of maternal newborn and child health services in Tanzania. The facilitators for maternal and perinatal death reviews were the prioritization of the maternal health agenda and the presence of active maternal death committees at the facility. In terms of barriers, there was perceived negligence among leaders, frustrations due to poor implementation of recommendations made by the committees, and poor record keeping. In terms of the facilitators for monitoring and evaluation, this study found the availability of a health management information system (HMIS), day-to-day implementation of ward rounds, and use of technologies. For the barriers to monitoring and evaluation, this study found the non-use of data for decision-making, ad hoc supervision, and the shortage of human resources for health and supply chain deficiencies.

### 4.1. Facilitators for Implementing Maternal and Perinatal Death Reviews

This study found that the prioritization of maternal health agenda by the government likely contributes to the smooth implementation of maternal and perinatal death reviews. This finding is consistent with findings from other studies [6,7]. The latter studies insisted that the maternal health agenda was not only the priority of the government but regarded as a human right. Having categorized it in that way, the government of Morocco organized actors such as public and private sectors, policymakers, civil society organizations, NGOs, and international organizations such as the UN to work together to curb the public health concerns resulting from maternal mortality. Likewise, a study conducted in Uganda [8] noted the same view that prioritization of the maternal health agenda has to feature in collaborative research projects by formulating relevant research questions for maternal and newborn health in villages. It also insisted on prioritizing research questions that address maternal and child health that are of public interest because they will be easily taken up by the Ministry of Health, academics, and funding agencies. This means that research on maternal and newborn health should reflect issues of local interest and not solely rely on international funding agencies.

### 4.2. Perceived Existence of Active Maternal and Perinatal Death Reviews

This study found that regular schedules for meetings and consistent follow-ups fostered the implementation of maternal and perinatal death reviews. This finding is consistent with a study by Kodan [9], who contended that the presence of active maternal death surveillance committees has improved the identification of maternal deaths since the year 2015. That study also indicates the existence of active maternal death review committees and initiated action plans for implementation, which were set as a working arm of the Ministry of Health to guide the implementation of their recommendations. Other studies have supported the views on the implementation of the action points recommended by the committees for further quality improvement and reduction of maternal deaths. For example, a study performed in the United States indicated that nurses and midwives can play a significant role in the activation of maternal review committees with their holistic approach and person-centered care philosophy [9,10]. Others have insisted on the multidisciplinary make-up, well equipped with a conceptual framework, contextual data, and evidence base needed to identify factors behind maternal deaths, causes of health inequity, indicators for community vital signs, and context within which women live and seek care. These are necessary for the maternal death review committees to identify novel and underlying factors for maternal mortality and propose novel prevention recommendations [5,12].

This finding is in line with that of Said [13], who argued that the review of maternal death audit is based on the comprehensive documentation of medical aspects that reveal a sequence of events leading to the death, identify gaps, and recommend measurable action plans for prevention. Other studies have shown some pessimism because not much is known about what works well, both where and why, in scaling up this accountability framework. This indicates that maternal death reviews are not adequately institutionalized and shift from facility to inform the wider health system [1,14,15].

### 4.3. Barriers to the Implementation of Maternal and Perinatal Death Reviews

This study indicated that there is perceived negligence among leaders who were reported not to adequately attend to their duties and responsibilities and, hence, compromised the implementation of maternal and perinatal death reviews. This finding is consistent with findings from other studies that highlighted issues of negligence, incompetence, and carelessness among health practitioners, which tend to undermine efforts to implement action points highlighted by maternal review committees [16,17,18]

There were also frustrations among providers of maternal care due to inadequate follow-up on the action points from maternal death review committees. This study found that there was poor implementation of the recommendations proposed by the review committees. This implies that if the recommendations from the review committees are not implemented, then there is continuity in a vicious cycle (little or no improvement at all). The study by Chrissman 2016 [19] pointed to similar findings by arguing that maternal death review committees’ recommendations suffer from clearly defined and attainable action points and from specific timelines, implementation methods, and individual responsibility assigned for implementation. This kind of finding supports the idea that a lack of follow-up together with inconsistencies characterizing recommendations proposed by the maternal review committees lead to frustrations in the health system and the dedicated efforts for quality improvement in maternal health care [20].

Furthermore, this study revealed that there are delays in conducting maternal and perinatal death reviews. This finding suggests that there are other delays apart from the inconsistencies in the implementation of the action points suggested by the maternal review committees. This signaled a huge health system constraint in the efforts for enhancing quality improvement in maternal health care services. This finding is consistent with findings from other studies [21,22], which highlighted that delays occur not only in conducting maternal death reviews but also in conducting caesarian sections [23], leading to increased maternal deaths. This implies that delays in conducting maternal death reviews tend to impair the improvement of the quality of maternal and newborn care [1,3].

### 4.4. Facilitators for Monitoring and Evaluation

This study pointed out that the availability of a health management information system at the facility makes it easier to track daily records of events and follow-ups on maternal health indicators. This finding implies that in the past experiences of using a paper-based system for recording keeping, there were setbacks such as incomplete data, which was resolved with the installation of HMIS. The study by Mboera [24] supports this finding by showing the importance of using HMIS when comparing performance in health service coverage and monitoring trends such as maternal mortality trends. It also shows that the presence of HMIS at a facility supports the systematic receiving and reporting of feedback and enhanced quality reports. Other studies have documented revolutions brought into the healthcare system in terms of enhancing data quality after the installation of the HMIS [25,26,27].

This study also underscored the viability of day-to-day implementation of ward rounds as a vital tool for enhancing M&E in maternal care services. This finding suggests that day-to-day ward rounds are vital for making follow-ups on the delivery of maternal and newborn care and the progress of the patient’s condition. This finding is consistent with findings from other studies [11,28] highlighting aspects of teamwork, sharing experiences, and ensuring constant monitoring of patients’ progress.

### 4.5. Barriers to Implementing M&E

This study showed that there were bottlenecks that impede the implementation of performance accountability mechanisms, M&E in particular.

In terms of a fragmented M&E system, the findings show that the M&E system is not working as required. This suggests that although there is a system for data collection, there is limited analysis to guide planning and decision-making—particularly for local improvement of quality of care. Eventually, the HMIS is reduced to a data collection exercise for transmission to the higher administrative levels within the health system [7]. Similar studies in sub-Saharan Africa show that districts are not equipped enough to use data generated by the HMIS for planning and decision-making [29,30].

Furthermore, this study revealed that ad hoc supportive supervision was a barrier to the implementation of M&E. The findings showed that regular supervision and monitoring of service provision was only performed during adverse events such as maternal deaths. This means that if the supervision is performed on an ad hoc basis, then the identification of risk factors for the occurrence of maternal deaths and implementation of prompt solutions is jeopardized. The aspect of supervision refers to the provision of guidance and feedback on matters of personal, professional, and health promotion, which involves the two-way process of guiding, joint problem solving, training, mentorship, helping, and encouraging healthcare workers to improve their performance in health service provision. At the regional and the council levels, supportive supervision is supposed to be performed quarterly under the leadership of the Regional Health Management Team (RHMT) and Council Health Management Team (CHMT), respectively [7]. Other studies have reported results consistent with these findings by highlighting issues of poor leadership, inadequate and unstructured supervision, and a lack of clear policy on supportive supervision, which tends to undermine efforts to implement an effective M&E process [7,31].

Finally, this study found an inadequate health workforce with the capacity for conducting M&E. This suggests that inadequate or lack of staff with knowledge and skills for M&E hinders the development and implementation of the M&E system. Consequently, the M&E of maternal and newborn care cannot be carried out effectively. This finding is consistent with findings from other studies [32,33] denoting that, for many years, Tanzania has been experiencing a serious shortage of human resources specialized in M&E. Other African countries are facing the same challenge, as highlighted in these studies [34,35].

### 4.6. Study Limitations

This study was conducted in one district hospital in the Pwani Region of Tanzania. This may cause a setback in making broad comparisons of the results presented in this study. Nonetheless, this limitation was offset by the depth of information and rigorous analysis using a qualitative approach. Moreover, the fact that health managers from each department/section involved in the provision of maternal health services were involved in the study as informants tended to enrich the quality of the data obtained from the responses to the intended questions. This may cause potential bias in the sense that the researcher’s personal/subjective feelings may influence the case study, but in the study process, the researcher used reflexivity to ensure that his past experiences and subjective feelings would not affect the research process or outcomes.

Furthermore, since the researchers work in a public health institution, it might have brought about social desirability [36,37,38] from the informants. However, this was minimized by performing a thorough debriefing of the subject matter of this study. We cultivated rapport by ensuring confidentiality, and we also used probes and other follow-up questions.

## 5. Conclusions

Most of the identified barriers to performance accountability mechanisms—maternal and perinatal death reviews together with monitoring and evaluation—are systemic and may account for limited effectiveness in the improvement of quality of care. The improvement of the quality of maternal and newborn care requires upholding the enablers and eliminating the barriers. Thus, based on the findings of this study, the District Health Management Teams (DHMTs) may institute appropriate corrective measures. For instance, maternal and perinatal death reviews may be strengthened by requiring the DHMTs to demand from leaders of health facilities comprehensive explanations about circumstances leading to every maternal death and measures to reduce the recurrence of similar incidents. The DHMTs may institute penalties against negligent leaders. For monitoring and evaluation, it is prudent that health facilities consider developing a fully functional M&E system that will have more than just collection and feeding of maternal and newborn data into the HMIS, but also perform the analysis and present the analytics to Management to guide planning, decision-making, evaluation of the extent of success, and adoption of corrective interventions.

## Figures and Tables

**Table 1 ijerph-20-06366-t001:** Summary of findings.

Facilitators for and Barriers to Implementing Maternal and Perinatal Death Reviews	
Facilitators for MPDR	Barriers to Implementing MPDR
Prioritization of the maternal health agenda by the government. Availability of active committees for the review of maternal and perinatal mortality.	Poorly functioning hospital system limits effective maternal death reviews. Perceived poor documentation. Frustrations due to problems identified during audits not being adequately implemented. Delays in conducting the reviews.
Facilitators for and Barriers to Implementing Monitoring and Evaluation	
Facilitators for M&E	Barriers to Implementing M&E
Availability of the health management information system (HMIS). Perceived informative online ordering systems for medical products. Day-to-day implementation of ward rounds.	Fragmented M&E system. Nonuse of data for decision-making. Supervision is performed on an ad hoc basis. Inadequate human resources for health skilled in M&E.

## Data Availability

Data for this study can be easily accessed through the office of the director of research and publication at Muhimbili University of Health and Allied Sciences in Tanzania at: https://www.muhas.ac.tz/ (accessed on 30 January 2023).

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
