# Peer review of "Facilitators for and Barriers to the Implementation of Performance Accountability Mechanisms for Quality Improvement in the Delivery of Maternal Health Services in a District Hospital in Pwani Region, Tanzania"

_ijerph, 2023, doi:10.3390/ijerph20146366_

Round 1

Reviewer 1 Report

The study is interesting and devoted on an underinvestigated country (Tanzani).

Hence, my overall opinion is positive even if some shortcoming should be adressed.

I'm a statistician, hence my observations are coherent with my expertise.

1. With respect to sample collection (p. 3, paragraph 2.3) the authors state that thirteen key informants are selected, but not information are provided on the methodology of selecetion of KI and neither on the the distribution of hospitals included in the analysis in the territory of the nation. I suggest to add more details on that point in order to be aware whethter the sample is representative or not.

2. The following point is not mandatory but suggested for improving the robusteness of the study; I would like to read summary statistics about the respondents of the study. I'm aware that the surveys are semi structured, but some key elements on the respondents should be revealed. 

Author Response

Response to Reviewer 1 Comments

Point 1: 1. With respect to sample collection (p. 3, paragraph 2.3) the authors state that thirteen key informants are selected, but no information is provided on the methodology of selection of KI and neither on the distribution of hospitals included in the analysis in the territory of the nation. I suggest to add more details on that point in order to be aware whether the sample is representative or not.

Response 1: Thanks for the comment. Thirteen key informants were selected purposively because they save as leaders/in-charges of the hospital’s sections. They include in-charge RCH – reproductive and child health, in-charge of the labour ward, matron, hospitals secretary, medical officer-in-charge, in-charge of the pharmacy, in-charge of the laboratory, in-charge of the theatre, the doctor in the labour ward, the doctor in the RCH, doctor in the theatre, anaesthesiologist, nurse in-charge of the complaints/opinions. Because of ethical reasons, we could not mention these titles in the paper because they carry direct identifiers of the informants. This was a qualitative study which focused on one district hospital as a case study. The aspect of representation does not strictly apply in qualitative studies.

Point 2: The following point is not mandatory but suggested for improving the robustness of the study; I would like to read summary statistics about the respondents of the study. I'm aware that the surveys are semi-structured but some key elements on the respondents should be revealed. 

Response 2: Thanks for the comment. The summary of the informants involved in the study were the health care workers working in different sections of the hospital who are directly or indirectly involved in reproductive and child health services. They were leaders in-charge of different sections/units like RCH, labour ward, laboratory, pharmacy, theatre, hospital secretary, medical officer in charge, matron, Doctor in charge of the RCH, labour ward, and anesthesiologist.

Reviewer 2 Report

Thank you so much for giving me the opportunity to read this article. Maternal and perinatal death reviews (MPDR)is one of the important systems to assess, and reduce maternal and newborn mortality in developing countries. Recently, the WHO launched the approach called Maternal and perinatal death surveillance and response (MPDSR)not only in Tanzania, but also other countries in African continent. Hence this topic is crucial.

This article is well-written, following the accurate methodologies, and analysis.

Here are my points which the authors need to amend or add more explanation.

Introduction

1.        It would be nice if the authors describe more about the similarly or difference between MPDSR, and MPDR. MDPR may be part of MPDSR, or being conducted separately?

https://www.who.int/teams/maternal-newborn-child-adolescent-health-and-ageing/maternal-health/maternal-and-perinatal-death-surveillance-and-response

2.        L60. Although performance accountability mechanisms exits…

The authors mentioned that there is such accountability mechanism (health political system)exist. however, there is no description what kind of the mechanism (implementation structures, and political system) exists in Tanzania. Please add more in detail about what kind of structures, who is in charge of implementing the M&E, and reporting in Tanzania. I would like to know overall-picture of the system (at central, district, and facility levels, and if the incidence happens, how the review should be conducted).

Methods

3.        Data analysis is conducted with accurate steps. If available, the authors can upload interview guide for reference (or the brief summary of questions)

Results

4.        Table 1

Table 1 shows the facilitating factors and barriers for M& E first, followed by MPDR. Hence description of results should be consistent in order. Or, if the aim focusing on the MPDR, Table 1 should be modified in order (1. MDPR, and 2. M&E).

5.        Perceived negligence among the leaders

This category mentions irresponsible leaders at the health facility. This problem is common in different approaches, and other countries. The authors need to add more explanation about who is in charge of the supervision, and why this happen in the context of Tanzania in this results, or in discussion part.

6.        Delays in conducting the review

The results are also interesting. However, the delay occur due to the fear of blame is not unclear why such happens. People might take action because they fear of blame. Please add more explanation.

7.        Inadequate human resources for health skilled in monitoring and evaluation skills

As for the findings, because the authors did not explain about the implementation system in introduction part, we could not understand the matter and hiding situation clearly. I would like to ask the authors who (at which level such as management level) should be in charge of the M&E. Also, if there is any eligible criteria for the post, the authors should describe or add some appendix in introduction part. Committee members

Discussions

8.        L478 ad-hoc supportive supervision

This sentence was not clear both in results, and the discussion. I would like to ask the authors, what supposed to be done for accurate supportive supervision (content, and frequency), and why there is such gap between the reality and the ideality.

9.        Study limitations

This study conducted in one district hospital in Pwani region. As the authors mentioned this is limitation, I would like the authors to add what kind of bias may exist in the findings considering the geographical and institutional characteristics.

10.      Typo and formatting error

The authors need to check the format, and English errors again before resubmitting the manuscript. 

Author Response

Response to Reviewer 2 Comments

Point 1: Introduction

  1. It would be nice if the authors describe more about the similarities and differences between MPDSR, and MPDR. MDPR may be part of MPDSR, or being conducted separately.

https://www.who.int/teams/maternal-newborn-child-adolescent-health-and-ageing/maternal-health/maternal-and-perinatal-death-surveillance-and-response

Response 1: Thanks for the comment, MPDR means reviewing circumstances surrounding the death to ascertain the causes of death. It has been in practice for a long time in different countries. It just described the causes of death without doing anything about it. On the other hand, MPDSR (Maternal and Perinatal Death Surveillance and Response involves constant surveillance to quantify deaths (Surveillance), then the review of circumstances to ascertain causes and contributing factors (MPDR, then making recommendations to address the causes (Response). It started in 2013. This emphasizes response which is making recommendations to address the causes and contributing factors. In essence, MPDR is part of MPDSR

 Point 2. L60. Although performance accountability mechanisms exist…

The authors mentioned that there is such accountability mechanism (health political system)exist. however, there is no description what kind of the mechanism (implementation structures, and political system) exists in Tanzania. Please add more in detail about what kind of structures, who is in charge of implementing the M&E, and reporting in Tanzania. I would like to know overall-picture of the system (at central, district, and facility levels, and if the incidence happens, how the review should be conducted).

Response 2: Thanks for the comment. Accountability mechanisms exist in the Tanzanian health system. In a study that explored accountability mechanisms existing in the district hospital, it showed that aspects like Opinion boxes, daily meetings (clinical meetings) ward rounds, complaint desks/office, Maternal and perinatal death reviews and OPRAS are some of the accountability mechanisms operating in Tanzania. It was also shown that the Health management team (HMT) is the entity responsible for making follow-ups in the implementation of these mechanisms. Specifically, the medical officer in –charge of the hospital, the matron and the hospital secretary are the key persons in charge of the implementation of M&E, lines 80 - 85

Methods

Point 3. Data analysis is conducted with accurate steps. If available, the authors can upload interview guide for reference (or the brief summary of questions)

Response 3: Thanks for the comment. The interview guide will be uploaded to the provided site on the journal webpage.

Results

POINT 4. Table 1

Table 1 shows the facilitating factors and barriers for M& E first, followed by MPDR. Hence description of results should be consistent in order. Or, if the aim focusing on the MPDR, Table 1 should be modified in order (1. MDPR, and 2. M&E).

Response 4: Thanks for the comment, the adjustments have been made in the document as suggested by the reviewer. Page 15 lines 194 to 195

POINT 5. Perceived negligence among the leaders

This category mentions irresponsible leaders at the health facility. This problem is common in different approaches, and other countries. The authors need to add more explanation about who is in charge of the supervision, and why this happen in the context of Tanzania in this results, or in discussion part.

Response 5: Thanks for the comment, the in-charge of supervision is at different levels depending on the hierarchy of authority. For instance, the medical officer in charge of the hospital is the overall charge of the supervision at the facility, and then the chain of command goes down to the specific sections like RCH, labour ward, theatre, and Laboratory which have specific personnel in charge of supervision. According to the feedback from the informants, negligence which may occur at any point in the hierarchy affects the whole system. One of the reasons pointed out was inadequate motivation, shortage of health workforce and finances. Line 252- 259

POINT 6. Delays in conducting the review

The results are also interesting. However, the delay occur due to the fear of blame is not unclear why such happens. People might take action because they fear of blame. Please add more explanation.

 Response 6: Thanks for the comment. Respondents alluded that delays in conducting reviews are an impediment to the implementation of MPDRs. It was also reported that delays happen because of fear of blame; leaders fear taking actions against the healthcare worker who were responsible for the deaths because they fear being blamed, this means that if responsible personnel perform reviews as early as required it will make it easy to uncover the reasons for deaths and propose actions against those responsible, but when they delay in conducting reviews the implementation becomes difficult since some information might be missing in the files, some persons might have been transferred to other sections. Thus it has become common that when matters are delayed it is difficult to come up with strict action points against those who were responsible. At the same time, leaders were reported to avoid blame since review reports come up with action points against the defaulters. Also, fear of blames was associated with inadequate supplies and human resource for health. Lines 296 - 306

POINT 7. Inadequate human resources for health skilled in monitoring and evaluation skills

As for the findings, because the authors did not explain about the implementation system in introduction part, we could not understand the matter and hiding situation clearly. I would like to ask the authors who (at which level such as management level) should be in charge of the M&E. Also, if there is any eligible criteria for the post, the authors should describe or add some appendix in introduction part. Committee members

Response 7: Thanks for the comment, Proper implementation of M&E is supposed to be carried out by well-trained and skilled experts in the subject matter. In Tanzania, the implementation of M&E is characterized by the use of HMIS which involves the use of registers documenting information about patient visits, diagnosis and treatment outcomes and health facilities. The aggregate summary of the forms and submitted to the district level to be entered into the computer software DHIS-2 unfortunately, the findings show that there are inadequate human resources for health skilled in M&E to perform respective tasks. The situation on the ground is that personnel at the facility responsible for performing M&E are not trained in that area, this suggests that the health management team members who are supposed to perform the tasks are not experts in M&E and thus resort to business as usual. Lines 397 - 402

Discussions

POINT 8. L478 ad-hoc supportive supervision

This sentence was not clear both in results, and the discussion. I would like to ask the authors, what supposed to be done for accurate supportive supervision (content, and frequency), and why there is such gap between the reality and the ideality.

Response 8: Thanks for the comment. The aspect of supervision refers to the provision of guidance and feedback on matters of personal, professional and health promotion, which involves the two-way process of guiding, joint problem solving, training, mentorship, and helping and encouraging healthcare workers to improve their performance in health service provision. At the regional and the council levels, supportive supervision is supposed to be done quarterly under the leadership of the RHMT – Regional Health Management Teams CHMT – Council Health Management Teams respectively. However, the findings show that supervision was only done on an ad-hoc basis. Particularly when there is an adverse event like maternal death. Lines 583 -589

POINT 9. Study limitations

This study conducted in one district hospital in Pwani region. As the authors mentioned this is limitation, I would like the authors to add what kind of bias may exist in the findings considering the geographical and institutional characteristics.

 Response 9: Thank you for the comment. It is true that the study took place in one district hospital which was regarded as a case study. The study aimed at getting in-depth information on the barriers and facilitators of implementing accountability mechanisms for quality improvement in the delivery of maternal health services. This may have potential for bias in the sense that the researcher’s personal/subjective feelings may influence the case study but in the study process the researcher employed reflexivity to ensure that his past experiences and subjective feelings wouldn’t affect the research process and outcomes. Lines 564 - 567

POINT 10. Typo and formatting error

The authors need to check the format, and English errors again before resubmitting the manuscript. 

 Response 10: Thanks for the comment and observation, the English language expert was consulted in the language formatting of the manuscript.

Reviewer 3 Report

Introduction

Lines 29-31: WHO has recently released MMR estimates for 2020, so the 2017 estimates given here should be updated

Line 39: typo – there’s an ‘i’ missing from ‘perinatal’

Line 38: A definition of ‘performance accountability’ should be provided, and a justification for why you decided to focus on this particular type of accountability. [I see there is a definition later in the introduction – but I think it should be defined the first time it is mentioned]

Line 39-40: This sentence implies that you decided a priori to focus on MPDRs and M&E. Is this the case? If yes, you should justify this decision, because other types of performance accountability mechanism exist. However, if these two mechanisms emerged from your study findings as the two main performance accountability mechanisms in use, then I think it’s too early to mention them here.

Line 42: Is 2016 the most recent available MMR estimate? If yes, might the authors consider using the 2020 modelled estimate from WHO?

Lines 42-44: please provide a citation for the systematic review that you mention

Line 44: typo – there’s an ‘i' missing from ‘arbitrarily’

Line 44: typo - it should be ‘shed light’, not ‘shade light’

Line 44: typo – it should be ‘specific’, not ‘spefic’

[There are numerous other types throughout the manuscript, I stopped recording them at this point]

Materials and methods

Line 92: I’m not sure ‘enforcement’ is the correct word here - perhaps ‘operation’ would be better?

Line 101: The data is nearly three years old. Can the authors provide reassurance to the reader that the findings are still applicable in the present day?

Lines 101-102: If I understand correctly, there are 9 DHs in the region, and the study was conducted in just one of the nine. Presumably you have not identified which one, so as to preserve anonymity? If so, I suggest you make it clear here that one of the nine listed hospitals was the study site. [Although I notice that later (line 166) you do name the hospital – if this was not an error, then I suggest you also indicate clearly here which hospital was the study site.]

Lines 105-106: Again, use of words like ‘enforcing to their subordinates’ implies that accountability mechanisms are imposed on the team at the hospital in a top-down way. Previous studies have found that staff ownership and a ‘no blame’ culture are key enablers for performance accountability mechanisms, and indeed you refer to this in the results (section 3.4.4). Perhaps the authors should reflect on whether the language used during the data collection and reporting was sensitive to this issue? Alternatively, it may simply be a language/translation issue – I’m conscious that I’m reading this in English and the study was conducted in Kiswahili.

Line 116: I would like to see more information in the manuscript about how the interview guide was developed: Who developed it? How did you decide which questions to include? How was it tested?

Line 144: What was the nature of the “prolonged engagement”? A 30-60 minute interview can’t really be described as “prolonged engagement”. Was there other engagement before or after the interview? If yes, this should be described.

Line 147: I don’t think you can claim that probing elicits “all the possible responses” – instead I think you could say that probing was used to encourage respondents to give detailed responses.

Line 171-172: Can you really claim that you’ve ensured anonymity? You’ve named the hospital so it would be possible for someone to work out the identity of most of the respondents. I recommend not naming the hospital in the manuscript.

Results

I like Table 1 -it’s a helpful summary. However, I suggest that you switch the order of the two types of mechanism, so that the order within the table matches the order within the narrative. Also, please ensure that the wording used in the table matches the section headings in the narrative – in some cases they are not exactly the same, which makes it difficult for the reader to navigate this section.

Line 197: The existence of facilitators that are specific to the individual hospital make me question the claims you make in the methods section about transferability of the findings. You mention this in the limitations, but perhaps you should also tone down the claims made in the methods section about transferability.

Section 3.4.1: Please clarify who are the “leaders” – many of the respondents could themselves be classed as leaders. Were they criticizing themselves, or other types of leader?

Section 3.4.1: I’m also puzzled by the complaint that they only turn up when there is a death. MPDRs should be triggered by a death, so why would they be expected to turn up at other times? I think perhaps you mean that they are not doing adequate follow-up, but if so, how is this different from Section 3.4.3?

Line 231: What is a “computer saver”? Do you mean “server”, or something else?

Section 3.4.2: Can you make it clearer why computerised records would be more consistent and complete than handwritten ones? I can see why they would be more accessible, but simply computerising the process won't guarantee improved completeness and consistency. This would happen only if the computerised forms were designed in a way to encourage complete and consistent reporting.

Section 3.4.4 – it should be “deceased”, not “diseased” or “deseeded”

Section 3.4.6: M&E is a very broad term. I think you should say more about what exactly is being monitored and evaluated. You mention “service delivery”, but what aspects of service delivery are covered by the M&E mechanism?

Section 3.4.7: I'm not sure you need a separate section for this. I suggest merging 3.4.6 and 3.4.7 into a single section, which is about computerised systems.

The section on barriers to M&E has insufficient detail about themes 1, 2 and 5 in Table 1.

Several of the themes reported in the M&E section relate not to M&E but to supportive supervision. I would class supportive supervision as an accountability mechanism in its own right, not just as something done as part of M&E efforts. Given this, I wonder if in fact your study is about three types of mechanism, not two?

Discussion

There is repetition in the discussion – you summarise the results at least twice.

If you decide to accept some of my recommendations to the results section, e.g. about organisation of themes, you’ll need to make similar adjustments to the discussion.

Author Response

REVIWER THREE COMMENTS

POINT 1. Lines 29-31: WHO has recently released MMR estimates for 2020, so the 2017 estimates given here should be updated

Response 1: Thanks for the comment, the new update has been inserted in the manuscript document. The current rate is 223 per 100,000 live births, this is the WHO estimate for the year 2020. Lines 38 - 39

POINT 2. Line 39: typo – there’s an ‘i’ missing from ‘perinatal’

Response2: Thanks for the observation, the correction has been made on line 39

POINT 3. Line 38: A definition of ‘performance accountability’ should be provided, and a justification for why you decided to focus on this particular type of accountability. [I see there is a definition later in the introduction – but I think it should be defined the first time it is mentioned]

Response 3: Thanks for the observation comment. A definition has been inserted in the manuscript document as suggested by the reviewer. Lines 42-43

POINT 4. Line 39-40: This sentence implies that you decided a priori to focus on MPDRs and M&E. Is this the case? If yes, you should justify this decision, because other types of performance accountability mechanism exist. However, if these two mechanisms emerged from your study findings as the two main performance accountability mechanisms in use, then I think it’s too early to mention them here.

Response 4. Thanks for the comment, it is true that performance accountability has several types other than maternal and perinatal death reviews, monitoring and evaluation, there are standards and bodies, health facility committees, together with professional norms. This paper focused on only MPDs and M&E because we employed a case study design that wanted to particularly dig deep into the two aspects. On the basis of the contextual reasons if enough emphasis can be invested in the two aspects it is very likely that quality improvement can be because if reviews and systematically performed and the associated recommendations are implemented alongside proper monitoring and evaluation of all the activities and procedures it very likely that quality improvement in maternal health can be improved/significant reduction of maternal mortality ratios and be realized. Lines 44 - 47

POINT 5. Line 42: Is 2016 the most recent available MMR estimate? If yes, might the authors consider using the 2020 modelled estimate from WHO?

Response 5. Thanks for the comment, the aspect of the most recent MMR for 2020 has been updated in the manuscript. Line 38-39.

POINT 6. Lines 42-44: please provide a citation for the systematic review that you mention

Response 6: thanks for the observation, the citation has been inserted in the document. Line 49

POINT 7. Line 44: typo – there’s an ‘i' missing from ‘arbitrarily’

Response 7. Thanks for the observation, the typo has been corrected. Line 51.

POINT 8. Line 44: typo - it should be ‘shed light’, not ‘shade light’

Response 8. Thanks for the observation. The correction has been made, line 51.

POINT 9. Line 44: typo – it should be ‘specific’, not ‘spefic’

[There are numerous other types throughout the manuscript, I stopped recording them at this point]

Response 9. Thanks for the observation. The corrections have been made throughout the manuscript. Line 52 and the rest of the pages.

Materials and methods

POINT 10. Line 92: I’m not sure ‘enforcement’ is the correct word here - perhaps ‘operation’ would be better?

Response 10. Thanks for the observation. The choice of the word enforcement is supported by the subject matter of accountability which is carried in the meaning of accountability itself. It means that enforcement is at the Centre of accountability.

POINT 11. Line 101: The data is nearly three years old. Can the authors provide reassurance to the reader that the findings are still applicable in the present day?

Response 11. Thanks for the observation. The data were collected almost three years ago but they are still applicable and reflect the present-day situation because, the data is within the range of five years – required time limit, but also during data analysis post visits were done to the field site for updates. Also, the phone numbers of the informants were recorded by the researcher, which call are made to ascertain the likelihood of any updates. 

POINT 12. Lines 101-102: If I understand correctly, there are 9 DHs in the region, and the study was conducted in just one of the nine. Presumably you have not identified which one, so as to preserve anonymity? If so, I suggest you make it clear here that one of the nine listed hospitals was the study site. [Although I notice that later (line 166) you do name the hospital – if this was not an error, then I suggest you also indicate clearly here which hospital was the study site.]

Response 12. Thanks for the comment. It is true that the study was conducted in one of the district hospitals, one of the nine hospitals in the region.

The name has been omitted for anonymity purpose, line 108.

POINT 13. Lines 105-106: Again, use of words like ‘enforcing to their subordinates’ implies that accountability mechanisms are imposed on the team at the hospital in a top-down way. Previous studies have found that staff ownership and a ‘no blame’ culture are key enablers for performance accountability mechanisms, and indeed you refer to this in the results (section 3.4.4). Perhaps the authors should reflect on whether the language used during the data collection and reporting was sensitive to this issue? Alternatively, it may simply be a language/translation issue – I’m conscious that I’m reading this in English and the study was conducted in Kiswahili.

Response 13. Thanks for the comment. We have recast the phrase ‘enforcing performance accountability to their subordinates’, it now reads “overseeing performance accountability in their respective units” which is less threatening. Lines 130-131

POINT 14. Line 116: I would like to see more information in the manuscript about how the interview guide was developed: Who developed it? How did you decide which questions to include? How was it tested?

Response 14: Thanks for the comment. The interview guide was developed in response to the research question: what are the facilitators and barriers to the implementation of performance accountability mechanisms in the delivery of quality maternal health services in a district hospital. Therefore, the guide had questions about existing accountability mechanisms, how they are implemented, factors enabling their implementation, and challenges facing the implementation of the identified mechanisms. The questions were developed by the first author and reviewed by other co-authors.  The tool was tested at Kisarawe district hospital, the questions were rephrased to fit the subject matter and more probes were added after the pilot study. Lines 142 -147 

POINT 15. Line 144: What was the nature of the “prolonged engagement”? A 30-60 minute interview can’t really be described as “prolonged engagement”. Was there other engagement before or after the interview? If yes, this should be described.

Response 15. Thanks for the comment. The nature of the prolonged engagement in the data collection process meant that the researcher used different techniques like paraphrasing the questions more probes during the interviews. Apart from the time spent during the interview other techniques like participant observation of the daily activities at the facility including how the suggestion boxes operate, and looking at how patients had access to the available phone number for reporting complaints. Also, because of the flexibility and comprehensiveness of the case study design, it enabled the researcher to broadly explore the barriers and potential facilitators to implementing performance accountability mechanisms for quality improvement in maternal health. Lines 174-179

POINT 16. Line 147: I don’t think you can claim that probing elicits “all the possible responses” – instead I think you could say that probing was used to encourage respondents to give detailed responses.

Response 16. Thanks for the comment. The corrections have been made to reflect the reviewer’s suggestion. Lines 155 to 156

POINT 17. Line 171-172: Can you really claim that you’ve ensured anonymity? You’ve named the hospital so it would be possible for someone to work out the identity of most of the respondents. I recommend not naming the hospital in the manuscript.

Response 17. Thanks for the comment. The name of the district hospital has been removed in the manuscript to maintain anonymity. Line 200

Results

POINT 18. I like Table 1 -it’s a helpful summary. However, I suggest that you switch the order of the two types of mechanism, so that the order within the table matches the order within the narrative. Also, please ensure that the wording used in the table matches the section headings in the narrative – in some cases they are not exactly the same, which makes it difficult for the reader to navigate this section.

Response 18. Thanks for the comment. The adjustments have been made in the manuscript to reflect the reviewers comment. Lines 211-212

POINT 19. Line 197: The existence of facilitators that are specific to the individual hospital make me question the claims you make in the methods section about transferability of the findings. You mention this in the limitations, but perhaps you should also tone down the claims made in the methods section about transferability.

Response 19. Thanks for the comment. I agreed with the comment. The adjustments have been made in the manuscript to reflect the reviewer’s comment. Lines 186-187

POINT 20. Section 3.4.1: Please clarify who are the “leaders” – many of the respondents could themselves be classed as leaders. Were they criticizing themselves, or other types of leader?

Response 20. Thanks for the comment. It is true that most of the informants interviewed were leaders. Most were heads of units or in charge of hospital sections like RCH and labour ward to mention a few. In the instances where they had concerns, they either had concerns about failure of their superiors at the facility or the leadership at the council or Ministry of Health, to fulfil their responsibilities towards implementing recommendations they had presented to them. The interviewees were the right informers because they are the ones who oversee day-to-day implementation of performance accountability mechanisms for quality improvement in maternal care, therefore the study sought to understand their perspectives on the barriers and facilitators of implementing accountability mechanisms.

POINT 21. Section 3.4.1: I’m also puzzled by the complaint that they only turn up when there is a death. MPDRs should be triggered by a death, so why would they be expected to turn up at other times? I think perhaps you mean that they are not doing adequate follow-up, but if so, how is this different from Section 3.4.3?

Response 21. Thanks for the comment. I agree with your assertion that MPDRs are trigged by deaths. 3.4.1 focuses on the negligence among leaders regarding the proper implementation of the reviews themselves and simply turning up to commence review when there is a new incident of maternal death. The findings in section 3.4.1 mean, as you have correctly said, leaders do not do adequate follow-up of actions for implementation in order to bring about improvement to halt a negative trend of maternal mortality. So have decided to merge two sections. The two are merged together. Lines 250 -284

POINT 22. Line 231: What is a “computer saver”? Do you mean “server”, or something else?

Response 22. Thanks for the comment. The typo has been corrected, it is computer server. Line 248

POINT 23. Section 3.4.2: Can you make it clearer why computerised records would be more consistent and complete than handwritten ones? I can see why they would be more accessible, but simply computerising the process won't guarantee improved completeness and consistency. This would happen only if the computerised forms were designed in a way to encourage complete and consistent reporting.

Response 23. Thanks for your comment. I agreed with your assertion that computerized records are more accessible and are guaranteed by designs. The concern from the informants was on the storage of the reports. The handwritten reports that can be precise and consistent by the storage system were noted to be a problem. Informants alluded that when records are kept on shelves sometimes they can be vandalized by unethical workers by deliberately removing pages containing information which may likely make them liable for repercussions

POINT 24. Section 3.4.4 – it should be “deceased”, not “diseased” or “deseeded”

Response 24. Thanks for the correction. It has been adjusted in the manuscript document. Line 276

POINT 25. Section 3.4.6: M&E is a very broad term. I think you should say more about what exactly is being monitored and evaluated. You mention “service delivery”, but what aspects of service delivery are covered by the M&E mechanism?

Response 25. Thanks for the comment. M&E as a mechanism for performance accountability, in the context of this study, implies the district hospital needs to have a hospital-based data collection to capture hospital status and health outcomes; and the monitor and evaluation performance by tacking indicators closely related to maternal and neonatal mortality. Such indicators include:

Impact indicators:

  • Number of maternal and newborn deaths.
  • Number of maternal deaths by cause: Postpartum haemorrhage Sepsis, Obstructed labour, Eclampsia, Unsafe abortion complications, HIV/AIDS (indirect) TB (indirect) Malaria (indirect)
  • Number of neonatal deaths (in the hospital)

Outcome indicators:

  • Percentage of deliveries in the hospital
  • Number of direct obstetric complications by cause: Postpartum hemorrhage Obstructed labour Eclampsia Unsafe abortion complications
  • Number of Cesarean sections performed
  • Number of newborns successfully resuscitated
  • Number of infants breastfed within one hour of birth

 Outputs indicators:

The number of EmONC signal functions performed, HIV tests conducted, and PMTCT (prevention of mother-to-child HIV transmission) services provided.

 Changes are inserted in the manuscript line. Lines 331 - 342

POINT 26. Section 3.4.7: I'm not sure you need a separate section for this. I suggest merging 3.4.6 and 3.4.7 into a single section, which is about computerized systems.

Response 26. Thanks for the comment. Section 3.4.6 is solely focusing on how DHIS-2 is useful in enhancing M&E capacity in the service delivery domain of the health system, while 3.4.7 focuses on the role of online system in enhancing the ordering of supply chain/medical products/medicines as a separate domain of the health system.

POINT 27. The section on barriers to M&E has insufficient detail about themes 1, 2 and 5 in Table 1.

Several of the themes reported in the M&E section relate not to M&E but to supportive supervision. I would class supportive supervision as an accountability mechanism in its own right, not just as something done as part of M&E efforts. Given this, I wonder if in fact your study is about three types of mechanism, not two?

Response 27. Thanks for the comment. True, supportive supervision is an accountability mechanism but is also applied to measure (evaluate) progress being made toward improvement (that is tracking indicators during supervision, making recommendations, and following up on implementation during the next supervisory visit). Supportive supervision is a facilitative approach to supervision that promotes mentorship, joint problem-solving and communication between supervisors and supervisees. Hence supportive supervision is implemented to improve routine program monitoring and evaluation (M&E). Thus the M&E and supportive supervision are inseparable(Nyamhanga, Frumence and Hurtig, 2021).

Discussion

POINT 28. There is repetition in the discussion – you summarise the results at least twice.

Response 28. Thanks for the comment, the repetition in the manuscript has been removed as suggested by the reviewer.

POINT 29. If you decide to accept some of my recommendations to the results section, e.g. about organization of themes, you’ll need to make similar adjustments to the discussion.

Response 29. Thanks for the comment, the recommendation from the reviewer has been taken into account in the manuscript.

Reviewer 4 Report

Dear authors, please consider the following comments in the hope of making you paper even better.

Please review your English ,e.g. line 60. Its difficult to understand in some places. Also many spaces between words in the text, e.g. line 67.

Please define accountability clearly. Please compare and contrast lines 37-39 with lines 52-59. 

Please explain if the case study is a representative case of what is happening in Tanzania. 

Are the results relevant to Tanzania only? Please clarify the contribution of your research. Why should people outside Tanzania read your research? What are the original findings?

Compare and contrast your findings with other similar research studies and the current body of knowledge.

Line 140, should it be bold? 

Best Regards.

Author Response

COMMENTS FROM REVIEWER 4

POINT 1. Please review your English, e.g. line 60. It’s difficult to understand in some places. Also many spaces between words in the text, e.g. line 67.

Response 1. Thanks for the comment. Thorough English language editing has been done throughout the manuscript.

POINT 2. Please define accountability clearly. Please compare and contrast lines 37-39 with lines 52-59. 

Response 2. Thanks for the comment. Line 37-39 explains the general selected definition of accountability, which involves to having the obligation to answer questions regarding decisions and/or actions and the imposition of sanctions for failure to comply and/or to engage in appropriate action. Then it continues to show that there are different types of accountability namely performance accountability, financial accountability together with social/political accountability.

Line 52-59 deeply explains performance accountability mechanisms which are the focus of this study. It means that performance accountability is further categorized into five types that are maternal and perinatal death reviews, professional norms, standards and bodies, committees’ tasks together with monitoring and evaluation. However, the findings of this particular study concentrated on only two of them namely maternal and perinatal death reviews together with M&E

POINT 3. Please explain if the case study is a representative case of what is happening in Tanzania. 

Response 3. Thanks for the comment. The aim of focusing on the case of one district hospital was guaranteed by the fact that the case study provides flexibility and comprehensiveness and, it enabled the researcher to broadly and deeply explore the barriers and potential facilitators to implementing performance accountability mechanisms for quality improvement in maternal health. In addition, the case study design helped to use various data collection methods like in-depth interviews, participant observation and document reviews. The case of the selected district hospital was supported by the fact that the 2012 census report for Tanzania showed that the Pwani region had 687 maternal deaths per 100,000 live births. Thus selecting one district in the region for gathering in-depth information can save the purpose of representation.

POINT 4. Are the results relevant to Tanzania only? Please clarify the contribution of your research. Why should people outside Tanzania read your research? What are the original findings?

Response 4. Thanks for the comment, findings are not only relevant for Tanzania, in the initial conceptualization of the study (defining the research problem) information from the global and regional level was taken into account. Also, the methods used in gathering information (findings) have been clearly described and can be replicated elsewhere outside Tanzania.

On one hand, People from outside Tanzania especially those in low and middle-income countries can take lessons from the findings of this study particularly the facilitators for implementation of performance accountability mechanisms which include; the use of technologies like online ordering of medical supplies, availability of DHIS-2 software for monitoring service delivery together with priority set by the government on maternal and child health agenda. On the other hand, findings from this study which explored the barriers to implementation of performance accountability mechanisms like fragmented M&E systems, non–use of data for decision making, leadership negligence and delays in conducting reviews; can be taken as lessons to leverage efforts taken in other contexts, especially in the low and middle-income settings. 

POINT 5. Compare and contrast your findings with other similar research studies and the current body of knowledge.

Response 5. Thank you for the comment. One of the findings regarding enablers of implementing MPDR was the prioritization the of maternal health agenda by the government. This finding is consistent with findings from other studies which emphasized that the maternal health agenda for Morocco was not only a priority but also regarded as a human right. In addition, the findings from a study conducted in Uganda documented similar attention to the prioritization of the maternal health agenda, where the government insisted it should feature in all research collaborations in the country. Other studies focused on barriers and facilitators of supportive supervision in maternal health services but this study focused on barriers and facilitators of two performance accountability mechanisms for quality improvement in the delivery of maternal health services.

The findings from this study are very pertinent and striking to the current body of knowledge as the world struggles to leverage every possible evidence-based intervention and therapy to significantly reduce maternal mortality ratios.

POINT 6. Line 140, should it be bold? 

Response 6. Thanks for the suggestion, the sub-heading has been bolded, line 157

Round 2

Reviewer 2 Report

Dear Authors,

Thank you so much for revising the manuscript.

I agreed with all of the amendments.

Please check the title mixing with capital letters and lower letters in the title. Also, I would suggest the authors check formatting (grammatical errors, abbreviations, and capital letters) again before publication.

Author Response

Round two; response to Reviewers' comments: Reviewer 2 Point 1. I agreed with all of the amendments Response 1. Thank you for the response Point 2. Please check the title mixing with capital letters and lower letters in the title. Response 2. Thanks for the comment; corrections have been made to remove the highlighted capital letters in the manuscript. Point 3. Would suggest the authors check formatting (grammatical errors, abbreviations, and capital letters) again before publication. Response 3. Thanks for the suggestion, revisions have been made around the highlighted aspects and corrections have been made